# Therapeutic Properties and Use of Extra Virgin Olive Oil in Clinical Nutrition: A Narrative Review and Literature Update

**DOI:** 10.3390/nu14071440

**Published:** 2022-03-31

**Authors:** Andrés Jiménez-Sánchez, Antonio Jesús Martínez-Ortega, Pablo Jesús Remón-Ruiz, Ana Piñar-Gutiérrez, José Luis Pereira-Cunill, Pedro Pablo García-Luna

**Affiliations:** 1Unidad de Gestión Clínica de Endocrinología y Nutrición, Hospital Universitario Virgen del Rocío, Avda. Manuel Siurot s/n, 41013 Seville, Spain; pabloj.remon.sspa@juntadeandalucia.es (P.J.R.-R.); ana.pinar.sspa@juntadeandalucia.es (A.P.-G.); josel.pereira.sspa@juntadeandalucia.es (J.L.P.-C.); 2Instituto de Biomedicina de Sevilla (IBiS), Hospital Universitario Virgen del Rocío/CSIC/Universidad de Sevilla, Avda. Manuel Siurot s/n, 41013 Seville, Spain; ajesus.martinez.sspa@juntadeandalucia.es; 3Unidad de Gestión Clínica de Endocrinología y Nutrición, Hospital Torrecárdenas, C. Hermandad de Donantes de Sangre, s/n, 04009 Almería, Spain

**Keywords:** olive oil, Mediterranean diet, enteral nutrition, parenteral nutrition, oleic acid, hydroxytyrosol, oleuropein, oleocanthal, diabetes mellitus, cancer

## Abstract

Extra virgin olive oil (EVOO) is a cornerstone of the Mediterranean diet (MedD). In this narrative review, we synthesize and illustrate the various characteristics and clinical applications of EVOO and its components—such as oleic acid, hydroxytyrosol, and oleuropein—in the field of clinical nutrition and dietetics. The evidence is split into diet therapy, oleic acid-based enteral nutrition formulations and oral supplementation formulations, oleic acid-based parenteral nutrition, and nutraceutical supplementation of minor components of EVOO. EVOO has diverse beneficial health properties, and current evidence supports the use of whole EVOO in diet therapy and the supplementation of its minor components to improve cardiovascular health, lipoprotein metabolism, and diabetes mellitus in clinical nutrition. Nevertheless, more intervention studies in humans are needed to chisel specific recommendations for its therapeutic use through different formulations in other specific diseases and clinical populations.

## 1. Introduction

Olive oil (OO) is the juice of the *Olea europaea* fruit. When it is obtained by physical processes and without any further treatment other than washing, decantation, centrifugation, and filtration, it is considered virgin olive oil. When VOO reaches the highest standards set by the EU Commission Regulation No 2568/91 [1] and the Trade Standard of the International Olive Council (IOC) [2], it can be classified as extra virgin olive oil (EVOO). Both regulations share the following requirements in chemical composition to define an OO as EVOO: a free acidity (expressed as percentage of free oleic acid) ≤0.8°%, a peroxide value ≤20 mEq O_2_/kg, UV spectrometry values that reflect minimal oxidation (K270 ≤ 0.22; K232 ≤ 2.50; ∆K ≤ 0.01), a favorable organoleptic assessment (median of 0 defects and median of “fruity” flavor >0), and a percentage of fatty acid ethyl esters (FAEEs) ≤30 mg/kg of EVOO (in crops harvested since 2015). The IOC also has a higher FAEE threshold (≤35 mg/kg of EVOO) and includes further qualifying criteria, such as ≤0.20% of moisture and volatile matter in the oil, ≤0.10% insoluble impurities in light petroleum, and low levels of trace metals (≤3.0 for iron and ≤0.1 for copper). If an OO complies with all the quality criteria to be defined as EVOO, then purity criteria must be met to maintain this certification, such as a ≤0.05 stigmadiene content (mg/kg of EVOO), small amounts of *trans*-isomers of unsaturated fatty acids (≤0.05 for the sum of C18:1 and ≤0.05 for the sum of C18:2 and C18:3), fatty acid content under specific thresholds (for myristic, linolenic, arachidic, eicosenoic, behenic, and lignoceric acids), ∆ECN42 (the difference between the theoretical and measured ECN42 by HPLC) ≤|0.2|, levels of 2-glyceryl-monopalmitate (depending on the palmitic acid percentage), certain requirements regarding sterols (a total sterol content ≥1000 mg/kg of EVOO, fixed maximum percentages for cholesterol, brassicasterol, campesterol, stigmasterol, ∆-7-stigmasterol, and the apparent sum of β-sitosterol, as well as a sum of erythrodiol and uvaol ≤4.5%), and total waxes (C42 + C44 + C46) ≤150 mg/kg of EVOO. The IOC has extra criteria, such as fixed ranges of other free fatty acids (palmitic, palmitoleic, heptadecanoic, heptadecenoic, stearic, and linoleic) and a higher threshold for eicosenoic acid (≤0.50 instead of ≤0.40) and total unsaponifiable matter (≤15 g/kg of EVOO). 

EVOO has been a cornerstone in the civilizations that have developed over millennia in the Mediterranean Basin. Since antiquity, peoples of the Mare Nostrum have been aware of its beneficial properties for health and have used this product in their gastronomy and pharmacopoeia [3,4,5]. The aim of this narrative review is to synthesize and illustrate the various characteristics, clinical applications, and mechanisms of action of EVOO and its components in the field of clinical nutrition and dietetics. More specifically, the following minor topics are discussed: diet therapy, enteral nutrition, oral nutritional supplementation, parenteral nutrition, and nutraceutical supplements. Our focus is on intervention studies in humans. When not available, evidence from other type of studies and/or animal models is reviewed. 

## 2. Materials and Methods

Each minor topic was individually explored in a comprehensive narrative literature review during October–November 2021 using PubMed and Web of Science search engines. MeSH terms were used when applicable, and different search strategies were undertaken combining the following keywords: *“**Mediterranean* *diet*”, *“extra virgin olive oil”*, *“olive oil”*, *“enteral nutrition”*, *“parenteral nutrition”*, *“oleic acid”*, *“tyrosol”*, *“**hydroxytyrosol”, “phenolic”*, *“oleuropein”*, *“oleocanthal”*, *“olive leaf”*, *“olive mill wastewater”*, *“antioxidant”*, *“inflammat*”*, *“glucose”*, *“cholesterol”*, *“metabolic syndrome”*, *“diabetes”*, *“obesity”*, *“lipoprotein”*, *“cancer”*, *“vascular”*, *“heart”*, *“osteoporosis”*, *“bone”*, *“prostat*”*, *“intestinal”*, *“neuro*”*, *“Alzheimer”*, *“fatty liver”*, *“elder*”*, *“microbio*”*, and *“arthritis”*. Our primary search was conducted in humans and intervention studies. If no evidence was available, then a secondary search was conducted selecting other type of studies and/or animal models. Finally, information was pooled and synthesized.

## 3. Results

### 3.1. Characteristics of EVOO

#### 3.1.1. Chemical Composition

The chemical composition of EVOO depends, among others, on the place of production, cultivar, and degree of ripeness of the olives, which can vary significantly [6,7,8,9,10,11]. From a chemical perspective, EVOO can be divided into three main groups of substances:
1.Saponifiable fraction. Accounts for 98.5–99.5% in total weight [6,7,8,9,12,13,14]. It is mainly made up of the following:
(1)Free fatty acids. The most frequent are palmitic (C16:0), palmitoleic (C16:1), stearic (C18:0), oleic (C18:1), linoleic (C18:2), and linolenic (C18:3) acids. Myristic (C14:0), heptadecanoic, and eicosanoic acids are also present in small amounts [6,7,8,9,12,13,14]. See Appendix A for further information on the percentages of fatty acids present in EVOO from the main cultivars in the Mediterranean Area [6,15,16,17,18,19,20,21,22,23,24,25,26].(2)Triglycerides (TAGs) account for 99% of the saponifiable fraction. The most prevalent (43.5%) is triolein, composed of three chains of acid. Other esterification possibilities are one palmitic acid in position *sn*-3 and two oleic acids in *sn*-1 and *sn*-2 positions (accounting for 18.4% of TAGs), as well as one linoleic acid in *sn*-2 position bordered by two oleic acids (6.8%) [6,7,8,9,12,13,14].(3)Partial acylglycerols (diacylglycerols, DAGs, or monoacylglycerols, MAGs). Their presence in EVOO is due to incomplete triacylglycerol biosynthesis or hydrolytic reactions. A good-quality oil should have no more than 1–2.8% DAGs, with higher values being considered an indication of poor processing or storage. The 1,2-DAGs—typical of freshly produced oil—gradually transform into the more stable 1,3-DAGs. Therefore, the 1,3-DAG/1,2-DAG ratio is considered a reliable parameter to determine the age of EVOO. MAGs are much less prevalent (less than 0.25%) and vary widely according to the production area: Spanish, Portuguese, Italian, or Greek OOs are richer in oleic-based MAGs, while Tunisian OOs are richer in palmitic- and linoleic-based MAGs [6,7,8,9,12,13,14].2.Unsaponifiable fraction. Constitutes 0.5–1.5% of EVOO. It can be grouped into the following:
(1)Tocopherols. The most abundant is α-tocopherol (150 to 250 mg / kg of EVOO). The optimal ratio of vitamin E to polyunsaturated fatty acids of 1.5–2.0. α-Tocopherol is able to neutralize free radicals and prevent peroxidation of membrane lipids and low-density lipoproteins (LDLs). β-tocopherol (15–20%) and γ-tocopherol (7–23%) are present in smaller proportions [6,7,8,9,12,13,14].(2)Phytosterols. The amount varies from 100 to 250 mg/100 g of EVOO. Sterols are categorized into 4,4-dimethylsterols or triterpenic alcohols (two methyl groups at the C-4 position), 4-monomethylsterols or methylsterols (one methyl group at the C-4 position), and 4-desmethylsterols or phytosterols (no methyl group), the latter being the main sterols in EVOO [6]. EVOO has many phytosterols, such as β-sitosterol, campesterol, ∆5-avenasterol, stigmasterol, cholesterol, cholesterol, brassicasterol, sitostanol, ergosterol, campestanol, ∆7-cholestenol, ∆7-avenasterol, ∆7-stigmasterol, ∆7-campesterol, ∆5,24-stigmastadienol, ∆5,23-stigmastadienol, ∆7,24-ergostadienol, ∆7,22-ergostadienol, 22,23-dihydrobrassicasterol, and 24-methylene-cholesterol [27]. The main one is β-sitosterol, accounting for 75–90% of total phytosterols in EVOO [28]. Different vegetable oils have different proportions of phytosterols, such as higher campesterol for seed oil, and Δ7-stigmastenol for sunflower and saffron oil [29]. The IOC and EU Commission define the sterol “fingerprint” of EVOO as that with >93.0% of apparent β-sitosterol, which is the sum of the percentages of β-sitosterol plus Δ5,23-stigmastadienol, sitostanol, Δ5-avenasterol, Δ5,24-stigmastadienol, and chlerosterol [1,2]. For example, EVOOs from Picholine and Sinopolese cultivars in Italy have shown high and steady amounts of β-sitosterol through different seasons [30]. On the other hand, these regulations also limit the relative proportion of erythrodiol and uvaol as 4.5% [1,2], as excessive amounts of these triterpenes may be a sign of adulteration with olive pomace oil or grape seed oil. For example, EVOOs from Manzanilla Cacereña in Spain are well under this limit [31].3.Other minor components are the following:
(1)Phenolic compounds. They are divided into phenolic acids (*p*-coumaric, gallic, vanillic, and caffeic acid, as well as others), flavonoids (luteolin, apigenin, and their derivatives), lignans (pinoresinol and 1-acetoxypinoresinol), isochromans (1-phenyl-6,7-dihydroxy-isochroman, and 1-(30-methoxy-40-hydroxy)phenyl-6,7-dihydroxy-isochroman), secoiridoids (demethyloleuropein, oleuropein, ligstroside, and their aglycone derivatives), and phenolic alcohols (tyrosol, hydroxytyrosol, and homovanillyl alcohol) [27]. Tyrosol, 5-hydroxytyrosol, and its elenoic acid ester—oleuropein—have potent antioxidant activity in vitro and in vivo, along with anti-inflammatory activity. Combining an in vivo and in vitro approach, it was demonstrated that oleocanthal (a secoiridoid derivative) is a potent activator of the transient receptor potential ankyrin 1 (TRPA1) ion channel expressed in trigeminal sensory neurons. When activated, TRPA1 creates a Ca^2+^-mediated cell depolarization interpreted as a pungent sensation that triggers a defensive cough response, to prevent the entrance of potentially harmful compounds to the airway. This effect takes place mainly in the pharynx—where the density of TRPA1 receptors doubles that of the tongue—and independently of the lipophilicity of the food matrix [32]. Smaller amounts of simpler phenolic compounds—such as caffeic, vanillic, and ferulic acids—have the role of protecting and enhancing the bioavailability of α-tocopherol [6,7,8,9,12,13,14,33,34]. See Appendix A for further information regarding antioxidants in the unsaponifiable fraction of EVOO from the main cultivars in the Mediterranean Area [17,19,20,21,22,23,25,35,36,37,38,39,40,41,42,43,44,45,46,47,48,49,50,51].(2)Chlorophyllic and carotenoid pigments. Present in small amounts, they color EVOO. Of note are β-carotene and lycopene. [6,7,8,9,12,13,14].(3)Volatile compounds. More than 100 volatile compounds are responsible for the characteristic odor of EVOO. These are small molecules (<300 Da) that vaporize at room temperature [52] and are present in EVOO’s headspace in higher concentrations than lower-quality OOs [53,54]. The major volatile compounds in EVOO are C6 and C5 alcohols and aldehydes formed from polyunsaturated fatty acids (PUFAs) through the lipoxygenase pathway. Although the former are present in higher quantities, the latter have lower detection thresholds in olfactory receptors [55]. 2(*E*)-Hexenal is the main C6 aldehyde in EVOO, giving “green” and “fruity” notes to EVOO, while C5 aldehydes and alcohols give pungent sensations [56] that may be appreciated by consumers [57]. The concentration of volatile compounds in EVOO depends on the cultivar and geographical area where the olive tree lives [58]; hence, EVOOs with different origins can be sorted by their characteristic profiles [59]. Using this approach, a study clustered 18 Italian cultivars into eight groups on the basis of their distinct levels of ethanol, 2-methyl propanol, pentanol, *cis*-2-penten-1-ol, *cis*-3-hexenol, and octanol [60]. Unripe fruit is associated with hexanal, *trans*-3-hexexnol, *cis*-3-hexen-1-ol, and *cis*-2-hexenol, associated with compelling sensory attributes such as “fruity”, “grassy”, and “pungent” [61]. Higher oxygenation [62] and temperatures around 20 °C during malaxation [63], as well as an adequate storage temperature [64] and minimization of storage time [62], maximize the amount and quality of EVOO’s volatiles.(4)Aliphatic alcohols or fatty alcohols (FAL). They are useful to classify different categories of OO in relation to the wax content of the product. Even-chain FALs (ECFALS) are more frequent than odd-chain FALs (OCFALs). Policosanol is the mixture of the most common ECFALs: docosanol (C22), tetracosanol (C24), hexacosanol (C26), and octacosanol (C28). Policosanol content in EVOO is regulated by the IOC and EU Commission [1,2]. According to previous studies, hexacosanol seems to be the most abundant FAL in Croatian [65], Greek [66], Italian [67], Spanish [68], and Tunisian [69] cultivars. The FAL content of EVOO depends mainly on the cultivar [67]. Both OCFAL and ECFAL quantities decrease with the increased ripening of the fruit, but there are differences between studies regarding the FAL most affected by ripening. It was hexacosanol in Italian cultivars [70] and pentacosanol in Croatian cultivars [65].(5)Alkanes and alkenes. These linear hydrocarbons occur naturally in EVOO as precursors during wax biosynthesis. Alkanes and alkenes are mostly odd-numbered, as they come from the loss of a carboxyl group in very-long-chain acids (VLCA), which are initially formed by the fatty-acid elongation complex in the endoplasmic reticulum (using palmitic and stearic acids as substrates) [71]. The alkane content and profile in EVOO depend on its cultivar [72], harvest time [73], and geographical region: the most frequent *n*-alkanes are *n*-C25 in continental Europe, *n*-C23 in north Africa and southern Italy, and *n*-C29 in the Iberian Peninsula [73,74]. This different geographical location may be related to meteorological conditions, as the intensity of precipitation is associated with larger alkanes and alkenes [75]. In addition to their usefulness in tracing the geographical origin and cultivar of EVOO, alkanes and alkenes are also useful for detecting counterfeits, because vegetable oils, as well as olive oils of different qualities, have distinctive fingerprints regarding these compounds. Alkanes are absorbed and then metabolized to FALs in the small intestine [72].

#### 3.1.2. Production of EVOO and Its Minor Components

The profile and concentration of compounds in the unsaponifiable fraction of EVOO depend on the cultivar [76], irrigation [77], mode of cultivation [47], and ripening of the fruit [78]. Cultivar, time, and growing conditions can modify the expression of the β-glucosidase, a main enzyme in the synthesis of secoiridoid derivatives in *Olea europaea* [79]. Malaxation temperature [46] and storage temperature [80,81] may also affect the phenolic content of EVOO. During storage, there is time-dependent conversion of their secoiridoids and derivatives (mainly oleacein and oleocanthal) into byproducts such as hydroxytyrosol and oleocanthalic acid. During this process, different cultivars and geographical areas may present different transformation patterns [82]. There is no actual consensus on the definition of EVOO with high phenolic content. Some authors have proposed EVOOs with >500 mg of polyphenols per kg of product [76]. Polyphenols and other minor components can be extracted from olive pomace, olive mill wastewater, and olive leaves, creating a circular agriculture system. The phenolic profile of these non-oil sources of polyphenols is distinctive. Amongst them, olive tree leaves have the highest yield of phenols due to their greater content of secoiridoids (mainly oleuropein) [83,84]. The major determinant of the phenolic yield of olive tree leaves is the type of cultivar [85]. Other factors are the color [86] and dryness of the leaves [87], as well as the geographical location of the olive tree and leaf collection time [88].

#### 3.1.3. Cooking with EVOO

Crude EVOO is the ideal food matrix for absorption of its hydroxytyrosol in the intestines, ensuring maximum concentrations in blood [89,90]. Among heat-based cooking methods, frying and steaming may preserve or enhance EVOO’s antioxidant capacity, while sautéing, pressure cooking, boiling, and microwaving may lead to significant reductions in its total phenolic content [91,92]. When boiling vegetables in a pressure cooker, phenolic compounds are transferred to cooking water from both EVOO and vegetables, with a maximum transfer in the case of eggplant, tomato, and pumpkin [92]. Therefore, incorporating this cooking water into the final recipe—instead of discarding it—may be of interest to the consumer. During boiling, secoiridoids are transformed into tyrosol, which is increased, while tocopherol content is lowered. Interestingly, hydroxytyrosol content may increase for the first hours when using EVOO of the Picual cultivar, while the intensity of loss of total phenolic content may be cultivar-dependent [93].

During frying, hydroxytyrosol and tyrosol may have the lowest thermal resistance, and deep frying and pan frying may affect the type of phenolic compounds lost during heating [94]. When frying with EVOO, temperature seems a much more important factor than time; approximately, its phenolic content can be decreased by 40% at 120 °C and 75% at 170 °C with 15–60 min of cooking [95], while its scavenging activity may be halved at approximately 180 °C after just 15 min [96]. During intensive heating (180–220 °C for 30–120 min), EVOO can degrade more than other vegetable oils such as pomace, soybean, and palm oil, developing higher levels of peroxides and free fatty acids, as well as undergoing the highest relative loss in phenolic content [97]. Nevertheless, EVOO is more resistant to oxidation [98] and may produce higher yields of tocopherols at 180 °C than other monounsaturated fatty acid (MUFA)-rich cooking oils [99]. In this regard, using OO to fry blue fish (another important component of theMedD) may transfer significant amounts of squalene to the food matrix and protect its ω-3 content better than sunflower oil [100]. In this process, the loss of hydroxytyrosol and its secoiridoids seems to be greater than the loss of tyrosol and its derivatives [101]. Interestingly, the loss of total phenolic content under frying seems to take place early during cooking and stabilize from 60 min of cooking [102]. Under these conditions, 2(*E*)-hexenal content is drastically reduced in a time- and temperature-dependent manner, while new and undesirable volatiles appear due to oleic acid oxidation, thus reducing its sensory properties [103]. 

In a head-to-head study, microwaving for 5 min at 500 W did not generate significant losses in total phenolic content, while baking at 180 °C for 45 was associated with an 11% loss. In any case, both methods were less detrimental to the phenolic content than boiling at 100 °C for 40–80 min or frying at 180 °C for 1–5 h [1]. Nevertheless, microwaving EVOO at 500 W for longer periods of time is not recommended, as it may generate quantitatively greater oxidation and loss of phenolic content [104], as well as faster flavor deterioration through the formation of acrolein. Nevertheless, these results are limited by certain experimental conditions (microwave and oven samples did not have the same volume) [105]. Although lipid oxidation takes place during baking, the substitution of margarine for EVOO in bakery products can prevent triglyceride oxidation and increase its shelf life [106] without substantially altering the sensory and physical properties of the food [107]. This may be due to a greater presence of antioxidant molecules in the final product. In this regard, oleuropein levels may be higher in starch matrices that have been baked instead of boiled, while having greater bio-accessibility than lipid-based carriers [90]. A study on Caco cells suggested there may not be a close correlation between the total phenolic content of a certain polyphenol-enriched bakery product and its ability to exert anti-inflammatory properties, as it depends on the different bio-accessibility and bioavailability of each of these compounds. This may depend on the source of phenolic compounds, the type of molecule, the characteristics of the fermentation process, and the selected time and temperature during baking [108]. 

Consuming EVOO with egg or whey protein [109] and fortifying foods with polyphenols or secoiridoids may be of interest to people with an aversion to the bitter taste of these compounds [110]. Liquid preparations with oleuropein in comparison with capsules have shown a higher plasma peak of oleuropein and a faster absorption of hydroxytyrosol metabolites [111], which could explain the greater therapeutic efficacy associated with this pharmaceutical preparation [112]. Polyphenol absorption occurs in proximal sections of the small intestine in humans [113]. Drugs can interfere with the absorption of β-sitosterol. Ezetimibe is an approved treatment for phytosterolemia [114] that inhibits β-sitosterol absorption either as monotherapy [115] or in combination with statins [116]. β-Sitosterol absorption increases with atorvastatin [117].

#### 3.1.4. Absorption and Metabolization of EVOO and Its Minor Components

Secoiridoid derivatives and phenolic acids are hydroxylated in the mouth, stomach, and duodenum, releasing phenolic alcohols (tyrosol and hydroxytyrosol) [45] This transformation can be enhanced in vivo in mice with the addition of the probiotic *Lactobacillus plantarum* [118]. Interestingly, the bio-accessibility (fraction of bioactive compounds freed from the food matrix and available for their intestinal absorption) of phenolic compounds may be not only dose-dependent, but also contingent upon the different stability of these compounds with regard to digestive tract conditions and the characteristics of the food matrix. In a study with in vitro enzymatic digestion of EVOOs from five different cultivars, Sevillana had the lowest total phenolic content (571 ± 1.6 nmol/g of EVOO) and the second lowest hydroxytyrosol content (6.11 ± 0.0 nmol / g of EVOO), yet the highest bio-accessibility index for total phenolic compounds (36%) and hydroxytyrosol (2452%) [45]. Regarding the absorption of olive leaf extracts, a study found a total phenolic content of 94 ± 2 mg/g of dry extract, with oleuropein as its major constituent (76.1 ± 0.8 mg/g of dry extract). In this case, oleuropein also presented the highest bio-accessibility index (109.86%) after 240 min of enzymatic digestion [119]. Further studies focusing on the bio-accessibility of polyphenols after enzymatic digestion of different *Olea europaea* products have been extensively reviewed elsewhere [120]. Oleuropein and hydroxytyrosol antioxidant activity seemed to be similar in an ex vivo study in humans [121], while both seemed to be more potent than homovanillic alcohol in vitro [122]. Hydroxytyrosol plasma concentrations diminish within approximately 1 h after EVOO ingestion, as they are transformed into glucuronidated forms, which have similar antioxidant potency to tyrosol [123]. In humans, the absorbed dietary tyrosol is metabolized to hydroxytyrosol and homovanillic alcohol in the liver by the CYP2A6 and CYP2D6 enzymes [124].

Although phytosterol and phytostanol absorption in humans is poor, ∆5-phytosterols (sitosterol and campesterol) may have higher absorption and lower turnover rates than 5α-phytostanols (sitostanol and campestanol) [125]. This absorption takes place using the Niemann-Pick 1C-like 1 transporter (NPC1L1), the same route used by cholesterol. Therefore, the inhibition of cholesterol absorption is competitive and dose-dependent. Afterward, phytosterols are then esterified within enterocytes and incorporated in chylomicrons to reach the liver. Finally, hepatocytes secrete these phytosterols into the bile through ABCG5/G8 transporters [126]. Phytosterol composition, the characteristics of its food matrix, clinical conditions that affect cholesterol metabolism such as type 2 diabetes mellitus (DM2), and single-nucleotide polymorphisms (SNPs) in NPC1L1 and apolipoprotein E (ApoE) may modify both their bioavailability and their efficacy [126,127].

#### 3.1.5. Relevant Effects of EVOO on Pathophysiological Processes

Although EVOO has a high concentration of MUFA (oleic acid) compared to other fatty acids, EVOO’s minor components appear to be responsible for the greatest number of its beneficial effects. There is abundant evidence about the mechanisms of action and the therapeutic potential or capacity of these minor components, mainly in animal models or ex vivo studies in humans. The following is a very brief summary of the most clinically relevant evidence on the effects of EVOO:Anti-inflammatory. Consumption of EVOO in humans has shown significant reductions in inflammatory and cell adhesion mediators such as, but not limited to, vascular cell adhesion molecule 1 (VCAM-1), intercellular adhesion molecule 1 (ICAM-1), interleukin 6 (IL-6), and C-reactive protein (CRP), both pre-prandially and postprandially. After consuming a meal rich in EVOO, reduced levels of thromboxane B2, leukotriene B 4, tumor necrosis factor-α (TNF-α) mRNA, arachidonic acid synthesis, and natural killer (NK) cell activity have been observed in ex vivo human studies. These effects could be mediated—among other mechanisms—by direct activation of peroxisome proliferator-activated receptors- α/γ (PPAR-α/γ), repression of nuclear factor kappa-light-chain-enhancer of activated B cells (NF-κB), inhibition of cyclooxygenases 1 and 2 (COX1-COX2), lipoxygenases, MAP kinases (MAPK), and Janus kinases / signal transducer and activator of transcription proteins (JAK/STAT) pathways [12,14,128,129,130].Coagulation. EVOO appears to inhibit platelet adhesion by decreasing the von Willebrand factor expression, and regular consumption of EVOO may also reduce factor VII activation and circulating plasminogen activator inhibitor-1 (PAI-1) levels. [12,13,128].Vasodilation. In murine models, triterpenes generate aortic vasodilation in a NO-dependent and COX-independent manner [131] that is endothelium-mediated [132]. EVOO’s minor components may also negatively regulate *ACE* and *NR1H2* genes [130].Improved endothelial health and lipoprotein composition. The combination of the previously described properties of EVOO allows it to improve endothelial function [12,14] and reduce the risk of atheromatous plaque development or complication [12,13,14,128,130]. Regular consumption of EVOO increases the high-density lipoprotein (HDL) fraction and the phenolic content of LDLs, improving their resistance to oxidation and, thus, reducing their atherogenic capacity. Although mixed, some evidence indicates that EVOO may also reduce abdominal adiposity [130].Control of cell proliferation. Phenolic compounds, such as oleocanthal, are able to act on the expression of genes that control the proliferation, apoptosis, and differentiation of cancer cells [13,33,34,133]. There is in vitro evidence showing that polyphenols in EVOO alter microRNAs in cancer cells, while hydroxytyrosol appears to exert strong antiproliferative effects in human colon adenocarcinoma cells [14].Microbiological properties. Oleuropein has in vitro antimicrobial activity against multiple Gram-positive and Gram-negative bacteria, mycoplasma, and viruses [134]. Some of these microorganisms are of particular interest, such as methicillin-resistant *Staphylococcus aureus* (MRSA) [135]. Oleuropein caused improved survival in an animal model of multidrug-resistant *Pseudomonas aeruginosa* sepsis, explained by ex vivo stimulation of phagocytosis and inhibition of proinflammatory cytokines [136]. Hydroxytyrosol has shown ex vivo and in vivo anti-HIV-1 activity in rabbits, with no associated toxicity [137]. EVOO consumption has been linked to increased proportions of *Clostridium* cluster *XIVa* and *Bifidobacterium*, *Parascardovia, Bacteroides fragilis*, and *Lactobacillus johnsonii*, as well as a reduction of the presence of *Lactobacillus animalis, Lactobacillus taiwanensis, Lactococcus* spp., *Proteobacteria, Deferribacteres*, and *Rikenella* [130].Bone homeostasis. In an animal model, both hydroxytyrosol and tyrosol appeared to have a positive effect on bone health [128]. In vitro studies in the human osteoblast cell line MG63 have shown differential effects of caffeic, ferulic and coumaric acid, apigenin and luteolin on gene expression [138,139]. The addition of different varieties of EVOO—with unique compositions in minor components—had divergent effects on the differentiation, antigenic expression, and antigenic capacity of this cell line [140].Cardiac function. In animal models, oleuropein protected in a dose-dependent fashion against loss of ejection fraction and cardiac output due to myocardial infarction at 24 h [141] and 5 weeks [142]. Oleuropein also protected in a redox-dependent manner against reperfusion injury in similar studies [143,144]. In animal models, oleuropein prevented myocardial remodeling in heart failure—by inhibiting the angiotensin-converting enzyme [145], in doxorubicin cardiomyopathy—by modulating nitric oxide synthases and other mechanisms [146], and in autoimmune myocarditis [147] and sepsis-induced myocardial injury—by NF-κB inhibition [148]. Oleuropein and hydroxytyrosol exert antiapoptotic effects on cardiomyocytes by reducing endoplasmic reticulum stress [149]. In an animal model of obesity, oleuropein and caffeic acid improved myocardiocyte energy yield without improving redox markers [150].Amyloid production. Oleuropein offered protection against amyloid deposition in a human cell line that mimics familial systemic amyloidosis [151].

### 3.2. Use of EVOO in Diet Therapy

The Mediterranean diet (MedD) is a dietary pattern typical of countries characterized by their abundance of olive trees and the extensive use of OO in their diets, being their main source of fat. Interest in the MedD began in the 1960s with the Seven Countries population-based study, which observed up to three times lower mortality due to heart disease in Mediterranean countries compared to those of northern Europe and the United States [152]. The composition of EVOO and its effects on multiple pathophysiological processes may synergize with other factors in this dietary pattern—which seem to be key to inducing the beneficial effects of MedD—such as an active lifestyle with regular physical exercise, high fiber intake, consumption of foods naturally rich in ω-3 fatty acids, and compliance with nutritional recommendations on micro and macronutrient composition, among others. The consumption of EVOO has shown beneficial effects in many different pathologies, all of which are related to its antioxidant, immunomodulatory, and inflammatory response-regulating effects. Below, we summarize the evidence for high-impact pathologies in the general population.

Cardiovascular function and health. The MUFA composition of EVOO has been associated with better peripheral insulin sensitivity and glucose tolerance, reduced exogenous insulin requirements, improved endothelial function, and antithrombotic properties [153]. Clinical trials have shown that the phenolic content of EVOO is responsible for its antihypertensive properties [154,155] and its improvement of fasting and postprandial lipid profile [153] in a dose-dependent manner [156] in different clinical populations, including DM2 [157]. Clinical trials with polyphenol-rich EVOO have also shown its ability to modify levels of antioxidants, proinflammatory proteins, biomarkers of endothelial dysfunction, and DNA and lipid oxidation in a dose-dependent manner, in various clinical populations [155,158,159,160,161,162,163]. In this regard, a meta-analysis of clinical trials—with a majority of healthy adults—confirmed that high phenolic EVOO (defined as ≥200 mg/kg of phenolic content) has the greatest impact in the reduction of oxidized LDL-cholesterol (HR: −0.68; 95% CI: −1.31, −0.04) [164]. The PREDIMED study has been a milestone in this topic. It was an unblinded clinical trial, designed to test the efficacy of MedD in the primary prevention of cardiovascular major events (a composite of myocardial infarction, stroke, and death from cardiovascular causes) in a high-risk ambulatory population. The treatment arm exposed to dietary training to increase adhesion to MedD, and four tablespoons (approximately 52 g) of EVOO per day had a 31% risk reduction (HR: 0.69; 95% CI: 0.53, 0.91) of cardiovascular major events [165]. An observational post-trial follow-up in the PREDIMED study found a 10% risk reduction (HR: 0.93; 95% CI: 085, 0.95) in cardiovascular major events for each 10 g of EVOO that participants consumed at baseline, independent of their compliance to the MedD and other confounding factors at that moment [166]. A meta-analysis of cohort studies attributed a reduction in cardiovascular mortality of 21% to MedD [167]. Participants in the NHANES and UPFS cohorts that consumed more than 7 g of OO per day had a 19% risk reduction (HR: 0.81; 95% CI: 0.75, 0.87) in comparison with those that never or seldom did [168]. The substitution of an equivalent dose of margarine or butter for a tablespoon of OO showed a 6% risk reduction (HR: 0.94; 95% CI: 0.90, 0.97) in the NIH-AARP Diet and Health Study cohort [169]. Population studies also associate EVOO consumption with reduced vascular stiffness [170] and attribute the reduction in cardiovascular mortality from MedD to hydroxytyrosol and its metabolites [171]. A clinical trial in healthy volunteers found that EVOO improved proteomic markers associated with coronary artery disease, while no differences were found for those associated with chronic kidney disease or diabetes [172].Lipoprotein metabolism. In 2011, the European Food Safety Authority (EFSA) issued a Scientific Opinion accepting the benefits of the daily intake of 5 mg of hydroxytyrosol and its derivatives from EVOO to prevent oxidation of LDL particles in the general population [173]. In randomized, blinded, and crossover clinical trials, the addition of polyphenol-rich EVOO was shown to reduce the expression of genes related to inflammation, atherogenesis, oxidation, metabolic syndrome, dyslipidemia, and DM2 in different clinical populations [158,174,175,176], as well as increasing the expression of genes involved in cholesterol efflux to HDL particles in people with hypertension [177]. Similarly, polyphenol-rich EVOO has the highest capacity to improve the number, proportions, and morphology of lipoproteins [178,179], due to the incorporation of ingested polyphenols into these particles [180]. Results from an animal model suggest that polyphenol supplementation and exercise may synergize to achieve a greater reduction of oxidized LDL particles [181]. On the other hand, an isocaloric high-MUFA (sourced from sunflower oil and margarine), high-fiber, and cholesterol-restricted diet was able to maintain apolipoprotein AI levels while providing the same decrease in apolipoprotein B100 as an isocaloric low-MUFA, high-fiber, and cholesterol-restricted diet in a clinical trial with people with dyslipidemia [182].Alterations in carbohydrate metabolism. MedD has been associated with weight loss and protection against metabolic syndrome in humans [156]. The consumption of EVOO in the general population, at the expense of its phenolic content, has been associated with a 40% relative risk reduction in the incidence of DM2 [183]. Metabolic control in people with DM2 may also benefit from the use of EVOO, by delaying the need for initiation of antidiabetic treatment [184] and reducing exogenous insulin requirements [153]. In pregnant women, consumption of EVOO has been associated with a 27% relative risk reduction of newly diagnosed gestational diabetes [185]. In people with type 1 diabetes mellitus (DM1), EVOO has been associated with an improvement in weight, waist circumference [186], lipid profile [187], and quality of life [188]. Lastly, polyphenol-rich EVOO has maintained insulin secretion and insulin sensitivity during a high-fat diet in animal models [189].Neurodegenerative diseases. Many neurodegenerative diseases share oxidative stress and neuroinflammation in their genesis and progression. Oleuropein, hydroxytyrosol, and oleocanthal are components of EVOO specifically related to the reduction in oxidative stress. Despite a suitable theoretical framework, there are not many studies that allow us to elucidate the effects of EVOO consumption on neurodegenerative diseases in humans [190]. In this regard, a long-term follow-up (mean of 6.5 years) of the PREDIMED clinical trial showed—in comparison to a low-fat diet—cognitive improvements in the EVOO supplementation and MedD dietary training group, which suggested a neuroprotective effect [191]. These findings are congruent with the results obtained in the NHANES and HPFS cohorts. Participants that consumed more than 7 g of OO per day showed a 29% lower risk of neurodegenerative disease mortality (HR: 0.71; 95% CI: 0.64–0.78), in comparison with those that never or seldom consumed OO [168].Cancer. An ex vivo study in humans showed how EVOO modifies the expression of oncogenesis-related genes [192]. In observational and experimental studies in humans, EVOO consumption was associated with an overall 31% risk reduction in the incidence of any type of malignancy, both within and outside the MedD setting. The highest risk reduction appeared in pancreatic, esophageal, and urinary cancer (54%), followed by breast cancer (33%) and gastrointestinal cancer (23%). This protection may be mediated by decreased expression of receptor tyrosine-protein kinase erbB-2 (HER-2), upregulation of repressor of HER-2 expression (PEA3), increased expression of cannabinoid receptor type 1 (CNR-1), inhibition of extracellular signal-regulated kinases 1/2 (ERK1/2) phosphorylation, or reduced expression of cyclin D1 [193].Gut microbiota. A 3 year sub-analysis of CORDIOPREV—a large clinical trial in people with metabolic syndrome and stable coronary heart disease—showed that the modifications of gut microbiota by EVOO supplementation (up to 138 g of EVOO per day, with >300 mg of phenolic compounds per kg of EVOO) and MedD differed according to sex [194]. In the MaPLE study—a crossover, randomized controlled clinical trial—a polyphenol-rich diet based on green tea, blood orange, Renetta apples, blueberries, and pomegranate juice (summing up to 724 mg of phenolic compounds per day) was able to improve the intestinal permeability and profile in the elderly [195]. In a small-sampled clinical trial, polyphenol-rich EVOO (500 mg of phenolic compounds per kg of EVOO) increased the proportion of IgA-coated bacteria—suggesting an increase in intestinal immunity [196]—while increasing the excretion of phenolic metabolites in feces without significantly changing the microbiome profile [197].

### 3.3. Oleic Acid-Based Formulations in Enteral Nutrition and Oral Supplementation

In this section, we summarize the evidence present in human studies on the use of lipids naturally present in olive oil or its major fatty component, oleic acid, in commercial enteral nutrition or oral nutritional supplementation formulations. Most of the available studies were conducted in the following clinical populations:Diabetes mellitus. Diabetes-specific enteral formulations (FEED) contain fructose, low-glycemic-index carbohydrates, and a higher proportion of fat than standard formulations (40–50%), mainly MUFAs in the form of oleic acid [198]. A large cohort found an initially neutral and subsequently protective effect of dietary oleic acid against the development of DM2 [199]. Most of the commercially available enteral nutrition and oral supplementation formulations source their oleic acid from sunflower and canola oil, as it is more cost-efficient than obtaining it from OO. A clinical trial in critically ill patients without diabetes compared two formulations with the same chemical composition—one sourcing its oleic acid from sunflower oil and the other from a mixture of sunflower oil and OO—finding no differences regarding the protective effect for this intervention on the development of stress hyperglycemia [200]. We did not find further specific data regarding OO-based enteral nutrition formulations in humans; thus, we focus on oleic-acid based FEEDs irrespective of their source of oleic acid. Meta-analyses and clinical trials have shown that FEED in critically ill and noncritically ill patients with DM2—whether administered orally or with an enteral access—can improve long-term glycemic control, decrease insulin requirements, and lower postprandial glycaemia [201,202]. Table 1 summarizes the main and latest clinical trials regarding oleic acid-based FEEDs in DM2 or stress hyperglycemia in different clinical settings. The improvement in glycemic control associated with FEEDs can be attributed to several mechanisms. On one hand, the high fat content delays gastric emptying, slowing glucose absorption [203]. This may lead to nausea, especially in patients with predisposing conditions [204]. On the other hand, oleic acid increases glucagon-like peptide 1 (GLP-1) and other neuropeptides via its metabolite 2-oleoyl glycerol [205]. An ex vivo study in humans associated the level of oleic acid in spinal fluid with improved respiratory quotient and glycemic response after oral glucose overload [206]. Ethnic origins may be taken into consideration when assessing the possible benefits of an oleic acid-based enteral formulation, as a clinical trial suggested that populations of African origin may have a lower insulin secretory response following an oleic acid-based high-fat diet [207].

2.Cardiovascular risk. The administration of FEEDs in clinical trials increased HDL-cholesterol and reduced LDL-cholesterol in noncritically ill patients with DM2 [203,226]. A meta-analysis found a significant improvement in HDL-cholesterol levels in people treated with FEED, although significance was lost in the sensitivity analysis [227]. Oleic acid was shown in a crossover clinical trial in people with DM2 to increase the concentration of the antioxidant enzyme paraoxonase 1 (PON1) in both chylomicrons and VLDL-cholesterol particles [228]. In a crossover clinical trial in healthy adults, a 3 week oral diet rich in oleic acid (comprising 28% of daily energy requirements) increased fasting respiratory quotient, reduced plasma LDL-cholesterol levels, tended to reduce fatty-acid oxidation, and significantly increased the expression of *insulin-induced gene-1* (*INSIG*-1), a molecule that modulates the activity of the sterol regulatory element-binding protein-2/1c (SREBP-2/1c) in women [229]. Of particular interest in DM2 and other clinical populations with high cardiovascular risk is to assess the long-term cardiovascular safety of an oleic acid-based formulation, although no specific studies are available regarding this topic. In a meta-analysis of cohort studies, MUFA intake did not present a lower risk for the development of cardiovascular events than saturated fat intake, but the authors stated that the mainly animal origin of MUFAs in the selected studies may have acted as a confounding factor [230]. MUFA intake has been associated with lower diastolic blood pressure [231]. Nevertheless, MUFA intake did not change surrogate biochemical markers of cardiovascular risk [232] or hemostasis parameters [233] in clinical trials, yet the oleic acid content of the erythrocyte membrane was inversely associated with biomarkers of inflammation [234].3.Obesity. Of particular interest in this clinical population are the possible effects of oleic acid on satiety and oral tolerance to oleic acid-based formulas. In clinical trials with healthy volunteers, oleic acid ingestion was shown to induce greater postprandial satiety [132] by increasing levels of the endocannabinoid oleylethanolamide, which was associated with a reduced caloric intake at the next meal [133]. However, this satiating capacity may be lost in overweight or obese individuals, due to a reduced hormonal and motor response to oleic acid intake because of their higher BMI and habitual dietary fat intake [134]. The oral taste detection threshold for oleic acid may be higher than for other fatty acids [135]. Adiponectin levels may be higher in TNF-α-stimulated adipocytes after a combination of hydroxytyrosol (1–10 μM) and oleic acid (10 μM) than in TNF-α-naïve adipocytes, as suggested in an in vitro study [235]. Therefore, creating formulas based on whole-EVOO instead of oleic acid might be of interest to potentiate this anti-obesogenic effect.4.Inflammatory bowel disease (IBD). Studies evaluating the role of oleic acid or MUFAs in the enteral nutrition received by patients with Crohn’s disease are scarce. In clinical trials, formulations rich in oleic acid and other MUFAs were less effective than those based on linoleic acid and PUFAs [236,237]. This is consistent with previous case–control studies that showed higher dietary intake of oleic acid and plasma levels of oleic acid in people with ulcerative colitis [238,239].5.Elder population. In a randomized clinical trial, the administration of an oral oleic acid-based formulation during medication rounds in elderly patients admitted to a geriatric ward was safe and well tolerated, and it increased appetite and daily caloric intake, while modestly improving lipid profile [240]. A similar study that orally supplemented an oleic acid-based formulation in elderly people with DM2 found a dose-dependent improvement in glycosylated hemoglobin [241]. To our knowledge, there are no published studies to date assessing the impact of an oleic acid-based enteral nutrition or MUFA-based enteral nutrition formula on sarcopenia in the elderly. In this respect, clinical trials in other clinical populations showed discordant results regarding the role of oleic acid-based formulations in the development of myosteatosis [242,243]. In our review of the available literature, we did not find intervention studies in humans that considered the role of oleic acid or MUFA-based enteral nutrition formulae on cognition in the elderly, although an oleic acid-based high-fat diet intervention was neutral for parameters of brain inflammation or impairment of memory tests in a crossover clinical trial in young people [244]. It is of interest to note that the anti-inflammatory and antiaging effect of MUFA consumption can be modulated by polymorphisms in the telomerase RNA component (TERC) in adults [245].

### 3.4. Oleic Acid-Based Formulations in Parenteral Nutrition

Intravenous lipid emulsions (ILEs) are a source of essential fatty acids and nonprotein calories used in the formulation of parenteral nutrition. Traditionally, soybean oil (SO) was used in their manufacturing. In the third generation of ILEs, OO was introduced in 80:20 ratio with SO, generating a formulation composed of 15% saturated fatty acids (SFAs), 65% MUFAs (especially oleic acid, which is an ω-9 fatty acid), and 20% essential PUFAs (linoleic acid and alpha-linoleic acid, which are ω-6 fatty acids) [246]. In the fourth generation of ILEs, the SMOFlipid^®^ 20% formulation (Fresenius Kabi Limited, Cheshire, UK) was introduced, which includes 25% OO in its composition, as well as 30% SO, 30% MCT, and 15% fish oil.

Evidence supports that OO-based formulations are associated with lower lipid peroxidation [129,247]. This results from the higher proportion of MUFAs than PUFAs in OO, which also results in these formulations not altering (or only slightly altering) inflammatory parameters when compared to traditional SO or SO emulsified with medium chain triglycerides (MCTs) [248,249]. In addition, MUFAs are associated with health benefits in the general population [250,251,252]. Another factor that may contribute to lower lipid peroxidation is the high α-tocopherol content of OO compared to SO, which contains higher concentrations of γ-tocopherol [253].

Third-generation and fourth-generation ILEs have lower ω-6:ω-3 fatty acid ratios than traditional ILEs [246]. Potential health benefits may derive from this, as ω-3 fatty acids have been associated with greater anti-inflammatory activity and vice versa [254]. ILEs containing OO could preserve the immune function, mainly through innate immunity [255,256,257]. In a study involving patients with advanced head and neck squamous carcinoma, the use of OO-based ILEs demonstrated an increase in natural and adaptive immunity, postulating that this could favorably influence their response to chemotherapy [258]. There are scarce data regarding infection in patients on parenteral nutrition as a function of the ILEs used; however, in the largest clinical trial to date, OO-based ILEs were associated with fewer infections when compared with soybean oil (SO)-based ILEs [259].

Another concern regarding the use of parenteral nutrition is its possible deleterious effect on the lipid profile of patients. OO-based ILEs may have beneficial effects on total cholesterol levels [129,260], and, although their relationship with blood TAGs is less clear, it has been shown in both children and adults that these formulas are safe and have little effect on the lipid profile when used long-term [248,261]. Regarding glucose metabolism, although little data exist, no differences have been found between OO-based and SO-based ILEs in adults [260].

With regard to liver function, there is no evidence that these OO-based ILEs alter liver function when used long-term [129] When comparing OO-based ILEs with other oil-based formulations, there are mixed results and differences in bilirubin, γ-glutamyl transferase (GGT), and alanine and aspartate aminotransferase (ALT, AST) levels that—while statistically significant—are not necessarily clinically significant, since these levels were within the normal ranges or within 1.5× upper limit of normal (ULN). Therefore, more studies are needed in this area [129].

Current results regarding clinical outcomes such as mortality and hospital or ICU stay did not differ between different types of ILEs in most studies, although two of them demonstrated a shorter duration of mechanical ventilation in patients receiving OO-based ILEs [262,263].

The maximum dose for OO-based ILEs is 2 g/kg of weight/day and should be reduced or discontinued if blood TAGs exceed 4.5 mmol/L [264] Regarding contraindications to the use of OO-based ILEs, they should not be used in patients with hypersensitivity to egg, OO, soy, peanuts, or any component thereof. In addition, OO-based ILEs formulations should be administered with caution in patients requiring anticoagulant therapy, as their lipids contain only 10–50 µg/L of vitamin K [250].

### 3.5. Supplementation of Nutraceuticals from EVOO

In this section, we summarize the studies that supplemented minor components of EVOO, either in isolation or within food matrices such as EVOO itself. Most of the available studies were conducted in the following clinical populations:Diabetes. Human clinical trials have demonstrated that oleuropein can improve postprandial glycemic response in healthy individuals [112], in overweight people without impaired carbohydrate metabolism [265], and in those with carbohydrate intolerance [266] and DM2 at the expense of increased GLP1 and insulin, by reducing oxidative stress through inhibition of NADPH oxidase 2 (Nox2) [267]. In a randomized, blinded clinical trial, olive leaf extract was also shown to improve postprandial glycemic response and metabolic control in people with DM2 [268]. In a prospective study in people with hydrocarbon intolerance, plasma β-sitosterol level acted as a protective factor against the development of DM2 [269]. Ex vivo studies in humans have indicated that oleuropein and hydroxytyrosol can inhibit α-amylase, α-glucosidase, sucrase, lipase, and glucose transporter 2 (GLUT-2) [112,270,271,272]. In animal models, oleanolic acid and luteolin have shown a superior ability to olive leaf extract in reducing postprandial blood glucose, all being significantly superior to placebo [266]. A meta-analysis of clinical trials found that phytosterols reduce insulin resistance, but not enough to generate clinically significant improvements in glucose metabolism [273]. This is backed by in vitro studies that demonstrated increased expression levels and activation of different components of the insulin receptor signaling pathway by phytosterols [274,275].Cardiovascular risk. In randomized, controlled, double-blind, and crossover clinical trials, administration of oleuropein and hydroxytyrosol [276]—as well as an olive leaf extract [277]—significantly reduced in vivo vascular stiffness and interleukin-8 production in prehypertensive individuals and healthy volunteers, respectively. Polyphenol administration in people with pre-diabetes or metabolic syndrome [278], as well as in overweight or obese people [279], improved biomarkers of endothelial dysfunction. This improvement in endothelial function was not associated with stimulation of angiogenesis [124]. In a double-blind, controlled, crossover clinical trial, hydroxytyrosol 7.5 mg b.i.d. increased antioxidant enzyme expression and reduced levels of oxidative stress markers [280]. A similar study—in which hydroxytyrosol 5 mg q.d and 25 mg q.d was administrated for 1 week—found no difference in levels of biomarkers of inflammatory and redox status, angiogenesis, and liver health, suggesting that this intervention requires a longer exposure time [281]. In vitro, both hydroxytyrosol and its metabolites lowered E-selectin, P-selectin, VCAM-1, ICAM-1, and MCP-1 levels in TNF-α-stimulated human endothelial cells, yet hydroxytyrosol metabolites interestingly exerted a greater effect than hydroxytyrosol at low concentrations (1 μM) [282]. A double-blind, controlled clinical trial in people with metabolic syndrome showed how supplementation of two minor components (chlorogenic acid and luteolin) improved intima-media thickness and flow-mediated dilation [283]. Regarding FALs, policosanol is a multifaceted drug with antiaggregant properties through direct inhibition of thromboxane A2 synthesis [284] and a lipid-lowering effect through the inhibition of cholesteryl ester transfer protein (CETP) activity, offering protection of LDL particles from oxidation [285,286]. In a double-blind, placebo-controlled, parallel clinical trial with ischemic heart disease patients, policosanol 5 mg b.i.d. with or without aspirin 125 mg q.d. reduced cardiac events and enhanced exercise tolerance [287]. Clinical trials have shown that policosanol 20 mg q.d. may have less adverse effects and better efficacy than low-intensity statins and low-dose aspirin in adults with dyslipidemia [288,289]. A meta-analysis calculated an LDL-cholesterol reduction of −23.7% for each 12 mg of policosanol per day [290], although other clinical trials found no beneficial effects in mild hypercholesterolemia [291]. Policosanol reduced LDL-cholesterol and triglyceride levels, increased HDL-cholesterol levels, and reduced systolic blood pressure in a double-blind, placebo-controlled, and parallel clinical trial with women with prehypertension [285]. In comparison to a high-fat diet, healthy C57BL/6 mice that were fed a high-fat diet with 0.5% policosanol for 18 weeks developed a better lipid profile, a reduction in brown and white inguinal adipose tissue, a distinctive microbiota profile, and enhanced lipolysis and thermogenesis [292], while, in a similar study, 60 mg/kg of either octacosanol or policosanol protected against visceral fat hyperplasia and liver steatosis [293]. Beneficial effects of phytosterols on LDL-cholesterol are well established, with a mean reduction of 8–10% [294] and significant reductions in ApoCII and ApoE [295]. An extensive review of clinical trials with phytosterols in cardiovascular risk reduction was published elsewhere [126]. There is no definitive evidence on the impact of dividing the daily phytosterol intake on cholesterol metabolism [296,297]; therefore, further studies are needed to make recommendations. The FDA has supported the claim that plant sterols and stanols—as part of a diet low in saturated fat and cholesterol—may reduce the risk of coronary heart disease (CHD), recommending an intake of 1.3 g or more per day of plant sterol esters (at least 0.65 g per serving) or 3.4 g or more per day of plant stanol esters (at least 1.7 g per serving) [298]. The phytosterol content of typical serving sizes of EVOO is well below the 0.65 g threshold. The association between phytosterols and CHD is complex, and new evidence supports that it may be different depending on the food matrix and demographics [299,300], clinical factors, and genetic traits [301,302]. Plant sterol oxidation products (POPs) and their consumption by nontarget population groups are also a concern. The EFSA has calculated that the approved maximum intake of 3 g of plant sterols per person per day may be deemed either safe or unsafe depending on its oxidation rate, which may depend on the cooking method, if the residual fat is discarded, and the characteristics of the food matrix [303]. Nevertheless, studies in animal models have shown that some POPs may have beneficial health properties, such as lipid-lowering properties (via PPAR-α activation, induction of cytochrome P450, and inhibition of enzymes involved in hepatic fatty-acid synthesis) [304].Obesity. There was a randomized, controlled, double-blind, and crossover clinical trial in which the addition of an olive leaf extract did not improve body composition or cardiovascular risk surrogates [265]. Oleuropein and hydroxytyrosol can reduce adipocyte hyperplasia and hypertrophy, by inhibition of adipogenesis-related genes in human cell lines [305,306] and stimulation of mitochondrial biogenesis-related genes in murine adipocytes [307]. When fed high-fat diets, EVOO-supplemented mice underwent lower weight gains [189], and they showed less metabolic, inflammatory, and microbiota disruption [308], as well as more modest subcutaneous and visceral fat development—in relation to a higher basal energy expenditure proportional to their weight—which could be explained by a higher expression of *Ucp* genes in muscle and visceral fat [309]. Regarding BMI reduction, a meta-analysis of clinical trials calculated statistically significant improvements for food enrichment with phytosterols and phytostanols in dyslipidemic, overweight, and obese subjects [310].Nonalcoholic fatty liver disease (NAFLD). To our knowledge, there have been no intervention studies in humans that supplement minor components in EVOO in patients with NAFLD. In humans, levels of sitosterols in the liver in adults with NAFLD have shown an inverse correlation with the degree of steatosis and lobular inflammation by liver biopsy [311] In animal models, liver damage induced by a high-fat diet was significantly reduced by supplementation with EVOO [312,313], EPA + DHA [314,315,316], DHA + EVOO [317] EPA + hydroxytyrosol [318,319,320,321], or hydroxytyrosol [322], showing beneficial effects on inflammation, lipid content, and hepatocyte gene expression. This evidence is mixed, as other animal models have not demonstrated a positive effect of EVOO and its phenolic compounds on the development of NAFLD [189] or its regression [323]. The beneficial effect of EVOO against NAFLD could be explained by mitochondrial protection—preventing the decrease in nitrated fatty-acid levels caused by a high-fat diet [324]—this ability being directly related to its phenolic content [325]. In addition, EVOO’s lipid profile is low in linoleic acid (ω-6), preventing a proinflammatory state due to an ω-6/ω-3 imbalance [326], with a high content of oleic acid (MUFA), transformed into oleylethanolamide and stimulating PPAR-α, and α-linolenic acid (PUFA), an ω-3 precursor that triggers antisteatotic, antioxidant, and anti-inflammatory responses in the liver via multiple molecular mechanisms [327].Cancer. A nutrigenomic study demonstrated ex vivo that oleuropein leaf extract can modify the expression of genes involved in oncogenesis in healthy humans [328]. There is an extensive review of direct antitumoral effects of oleuropein and hydroxytyrosol in in vitro and ex vivo models [329], which are linked to in vitro evidence of the intratumoral transformation of oleuropein into hydroxytyrosol [330]. Pleiotropic effects in vitro and ex vivo resulting from combinations of oleuropein, hydroxytyrosol, and other minor components with chemotherapeutics and immunomodulators, leading to drug synergism or a reduction in undesirable effects, were thoroughly reviewed elsewhere [33]. An extensive review of the effects of EVOO and its minor components in preclinical models of colon cancer is also available [331]. Oleuropein-rich EVOO stimulated in vivo apoptosis of colon cancer cells and peritoneal macrophages in an animal model [332], and a much more potent nanoencapsulated microemulsion of oleuropein was recently developed and tested in an animal model [333]. While the addition of oleic acid induced DNA synthesis and growth of the human colorectal adenocarcinoma cell line Caco-2, the subsequent addition of EVOO’s minor components prevented this effect, even inducing apoptosis in the case of hydroxytyrosol and maslinic acid [330]. Regarding mechanisms of action, oleuropein was able to decrease cell number and motility in the human melanoma cell line A375—decreasing their glycolytic capacity (Warburg effect)—at the expense of glucose transporter 1 (GLUT-1), pyruvate kinase M2 (PKM2), and monocarboxylate transporter 4 (MCT4) inhibition [334].Neurological diseases. In a randomized clinical trial, an oral palmitoylethanolamide and luteolin extract improved the detection threshold, discrimination ability, and smell identification in people with COVID-19 hyposmia [335]. Nevertheless, oral luteolin supplementation in a quasi-experimental study was not superior to placebo at reducing symptoms of Gulf War Illness, a neuroinflammatory condition [336]. Oleuropein has demonstrated neuroprotective effects in animal models of cerebral ischemia, cerebral reperfusion injury, ageing, Alzheimer’s disease, Huntington’s disease, multiple sclerosis, Parkinson’s disease, peripheral neuropathy, spinal cord injury, and epilepsy [337,338]. With respect to Alzheimer’s disease, oleocanthal was shown to increase neuronal clearance of β-amyloid by increasing P-glycoprotein (P-gp) and LDL receptor-related protein-1 (LRP1) levels in mouse brains [339], while restoring the blood–brain barrier function and reducing neuroinflammation through inhibition of PYD domain-containing protein 3 (NLRP3) inflammasome and inducing autophagy through activation of the AMP-activated protein kinase/Unc-51-like autophagy activating kinase 1 (AMPK/ULK1) pathway [340].IBD. In contrast to its lipid profile, the minority components of EVOO may have a beneficial effect on IBD, although we found no human intervention studies. Ex vivo studies in humans have pointed to the ability of oleuropein to reduce inflammatory infiltrate and inflammatory markers, improving tissue architecture [341,342]. In colitis animal models, oleuropein supplementation decreased mortality and improved histopathology by increasing antioxidant enzyme levels and Bcl2 expression while reducing proinflammatory cytokine and Bax expression [343], while EVOO supplementation improved rectal bleeding [344] and the visceromotor response to mechanical distension [345]. These beneficial effects may be enhanced by delivering oleuropein via lipid nanoparticles orally [346]. To gain further insight into this topic, we point the reader to an excellent review [347].Benign prostatic hypertrophy (BPH). In three randomized, double-blind, placebo-controlled clinical trials, β-sitosterol demonstrated superiority in improving self-reported somatic and sexual symptoms, quality of life, peak urine flow, post-void residual volume, and plasma free testosterone levels [127,348,349]. As possible explanatory mechanisms, this molecule reduced 5α-reductase activity and prostate volume in a hamster model of BPH [350], while unfractionated olive oil reduced prostate volume in a murine model of BPH, at the expense of decreasing prostatic testosterone (T) and dihydrotestosterone (DHT) concentrations [351].Osteoporosis. In a randomized, double-blind, 12 month clinical trial of postmenopausal women with mild osteoporosis and no drugs or diseases affecting phospho-calcium metabolism, the addition of an oral olive leaf extract containing 40% oleuropein to 1000 mg of calcium carbonate q.d maintained bone mineral density levels by DEXA and increased osteocalcin in a safe and well-tolerated manner [352]. An ex vivo study in humans using oleuropein [353] and in vivo studies in murine models of osteoporosis using EVOO and vitamin D3 [354], oleuropein [355,356], and hydroxytyrosol [357] have shown an anti-inflammatory effect—inhibiting osteoclastic activity and reducing trabecular bone loss—while in some cases also stimulating osteoblastic activity. An excellent review of previous studies indicated that EVOO’s bone protection is probably mediated by its anti-inflammatory capacity [358].Rheumatoid arthritis (RA). In a pilot intervention study in newly diagnosed patients, the addition of 35 mg oleuropein to methotrexate generated the maximum reduction in markers of cell damage, oxidative stress, and inflammation at 6 weeks [359]. In a randomized clinical trial with a mean duration of approximately 7 years, the addition of an olive leaf, fig leaf, and fruit extract to methotrexate improved patient-perceived health status without changing disease activity scales by 16 weeks [360]. In a randomized, crossover clinical trial, the addition of EVOO and fish oil to disease-modifying drugs (DMARDs) showed improved patient-perceived health status, increased functionality, and lowered rheumatoid factor levels at 24 weeks compared to the addition of fish oil to DMARDs [218]. All these interventions were safe and well tolerated. In the TOMORROW cohort, MUFA intake was significantly lower in RA patients, while MUFA intake tended to be an independent predictor of disease remission [361]. In a cross-sectional study, the consumption of olive products in the usual diet of people with RA showed a good correlation and explanation of variance in principal component analysis with respect to their calculated amount of dietetical plant-derived PUFAs. However, the amount of PUFAs was most strongly related to the consumption of walnuts and linoleic and alpha-linolenic acids, acting as possible confounding factors. In this study, consumption of PUFAs was associated with a lower degree of disease activity according to Disease Activity Score on 28 joints (DAS28), independently of antibody positivity and with higher protection in more severe or chronic forms of the disease [362].

## 4. Discussion

EVOO has a pivotal role in the MedD. With its well-balanced mixture of oleic acid and minor components, the culinary use of EVOO in the general population may contribute to the daily requirements of essential fatty acids and antioxidant body supply. Likewise, whole EVOO, its lipids, or minor components can be supplemented—in a safe, effective, and well-tolerated manner—in fortified foods, oral, enteral, and intravenous pharmacological formulations in clinical nutrition. Furthermore, diet therapy with EVOO and the nutraceutical supplementation of its minor components currently have a very scientifically sound rationale after showing multiple beneficial effects—in both intervention and mechanistic studies—in important conditions for public health, such as cardiovascular health, lipoprotein metabolism, and diabetes mellitus. Currently, promising therapeutic evidence—albeit in different stages of development, and too early to recommend specific EVOO-based dietary or nutraceutical interventions—is available for conditions such as obesity, NAFLD, neurodegenerative diseases, osteoporosis, cancer, and inflammatory diseases, amongst others. Regarding EVOO-based formulations in clinical nutrition, oleic acid-based ILEs already are well established parenteral formulations with widespread use in the clinical setting. To our knowledge, both oral and enteral oleic acid-based formulas may facilitate metabolic control, although adequately blinded, crossover clinical trials with a high number of participants and an intention-to-treat analysis are needed to further evaluate their safety and efficacy in specific situations.

Considering the aforementioned and possible future lines of research, it would be of high interest to develop intervention trials to further evaluate the effect of EVOO and its minor components on hard endpoints in high-impact clinical conditions such as heart disease, NAFLD, IBD, bacterial infections, or HIV. It would be of particular interest to assess the long-term cardiovascular safety and effects on hemostatic and inflammatory parameters of oleic acid-based formulations for enteral nutrition or oral supplementation—especially with minor components from EVOO—in groups with high cardiovascular risk. Clinical nutrition could also benefit from further studies comparing liver function under different OO-based ILEs in a randomized fashion in different clinical populations. Lastly, sarcopenia and cognition in the elderly are ever-developing fields that could benefit from well-planned studies assessing the impact of oleic acid-based oral and enteral nutrition formulations, as well as clinical trials designed to further test the efficacy and tolerability of EVOO’s minor components in age-related conditions such as osteoporosis, cancer, and dementia.

## 5. Conclusions

EVOO has diverse beneficial health properties and a well-established role in clinical nutrition. More specifically, the current evidence supports the use of whole EVOO in diet therapy and the supplementation of EVOO’s minor components to improve cardiovascular health, lipoprotein metabolism, and diabetes mellitus, as well as enough evidence for the use—when available—of oleic acid-based ILEs in parenteral nutrition. EVOO-based nutritional interventions have thus far shown very promising results in other high-impact clinical conditions, but more intervention studies in humans are needed in this case to develop specific recommendations and grant new therapeutic uses of EVOO through different formulations and in specific clinical populations.

## Figures and Tables

**Table 1 nutrients-14-01440-t001:** Characteristics of the latest clinical trials that used MUFA-based FEEDs. Modified from [201,202,208].

Author	Design	ITT	Setting	Group Characteristics	Administration	Time (Max)	I-Formula	I-Fat (%)	I-MUFA (%)	C-Formula	C-Fat (%)	C-MUFA (%)	Improvements
Lansink (2017) [209]	Double-blind, crossover	Yes	Ambulatory	Intervention and control groups DM2 (*n* = 24): 64.6 ± 10.7 years	Nasogastric tube, continuous feeding	4 h	Nutrison Advanced Diason Energy HP	46.4	27.6	Nutrison Energy Multi Fibre	34.4	19.4	Postprandial IS Postprandial BG
Gulati (2015) [203]	Open-label, crossover	No	Ambulatory	Intervention (*n* = 22), DM2, 45.1 ± 6.9 years; Control (*n* = 18), DM2, 47.7 ± 6.8 years	Oral	4 h	Nutren Diabetes	36.8	27.0	Isocaloric diet	NR	NR	Postprandial IS Postprandial BG
Alish (2010) [210]	Double-blind, crossover	No	Ambulatory	Intervention and control groups (*n* = 18): HbA1c 6.7% ± 0.2%, 63.1 ± 1.9 years	Oral	4 h	Glucerna 1.2	45.0	27.7	“Standard formula”	29.0	10.4	Postprandial BG
Pohl (2009) [211]	Double-blind, parallel	Yes	Neurological rehab Dept., nursing home, medical ward	Intervention (*n* = 48): DM2, HbA1c ≥ 7.0%, 74 (44–91) years; Control (*n* = 49): healthy, 69 (53–86) years	Nasogastric tube, continuous feeding	84 days	Diben	45.0	32.2	Isoenergetic, isonitrogenous enteral formula	30.0	17.0	Insulin requirements Fasting BG
Vaisman (2009) [212]	Double-blind, parallel	No	Ambulatory	Intervention (*n* = 12): HbA1c 7.0% ± 0.7%, 73.0 ± 14.7 years; Control (*n* = 13): HbA1c 7.3% ± 0.8%, 79.2 ± 10.4 years	Nasogastric tube, bolus feeding	12 weeks	Nutrison Advanced Diason	38.0	26.0	“Standard formula”	30.0	14.0	Postprandial BG HbA1c HDL-cholesterol
Vanschoonbeek (2009) [213]	Double-blind, crossover	No	Ambulatory	3 intervention and control groups (*n* = 15): DM2, 63.0 ± 1.0 years, HbA1c 7.3% ± 0.2%	Oral	4 h	Glucerna/Nutricia formula 1/Nutricia formula 2	50.0	34.8	Isosource Fiber	30.0	12.6	Postprandial IS Postprandial BG
Magnoni (2008) [214]	Double-blind, parallel	Yes	Ambulatory	Intervention and control groups (*n* = 39): DM2, HbA1c 6.5–8.5%, 57.5 ± 1.5 years	Oral	84 days	Diasip	49.0	34.2	“Standard formula”	30.0	17.1	Postprandial BG
Voss (2008) [215]	Double-blind, crossover	No	Ambulatory	2 intervention and control groups DM2 (*n*= 48): 56 ± 1.4 years	Oral	4 h	Diabetes-specific formula/Slowly digested carbohydrate formula	49.0/49.0	32.0/32.2	“Standard formula”	29.0	15.0	Postprandial IS Postprandial BG
Yokoyama (2008) [216]	Open-label, crossover	No	Ambulatory	Intervention: DM2 (*n* = 10), HbA1c 6.6% ± 0.7%, 58.6 ± 7.7 years; Control: healthy (*n* = 12), 20.8 ± 1.2 years	Oral	7 days	Glucerna	49.3	34.3	Enrich-SF (Meiji Milk Co.)	30.8	8.2	Lower AUC insulin and plasma glucose at 180 min
Hofman (2007) [217]	Double-blind, crossover	No	Ambulatory	2 intervention and control groups (*n* = 11): DM2, HbA1c 7.0% ± 1.2%, 63 ± 9.4 years	Nasogastric tube (continuous feeding)	6 h	Diason/“Test product”	38.0 /49.0	26.1/34.2	Nutrison Standard Multifiber	35.0	21.0	Postprandial IS Postprandial BG
Leon-Sanz (2005) [204]	Open-label, parallel	No	Medical ward	Intervention (*n* = 36) DM2, 7.7% ± 1.7 %, 73.9 ± 9.6 years; Control (*n* = 27), 7.8% ± 1.9%, 70.6 ± 8.7 years	Nasogastric tube (continuous feeding)	13 days	Glucerna 1.0	50.0	34.2	Precitene	31.0	9.4	Diarrhea, nausea
Pohl (2005) [218]	Double-blind, parallel	Yes	Neurological rehab Dept., nursing home	Intervention (*n* = 35): DM2, HbA1c ≥ 7.0%, 71 (42–86) years; Control (*n* = 37): Healthy, 72.0 (51–87) years	Nasogastric tube and gastrostomy, continuous feeding	84 days	Diben	45.0	32.2	Isoenergetic, isonitrogenous enteral formula	30.0	17.0	Insulin requirements Fasting BG
Doola (2019) [219]	Double-blind, parallel	No	ICU	Intervention (*n* = 21): 62.4 ± 9.9 years; Control (*n* = 20): 62.2 ± 11.3 years	Nasogastric or orogastric tube (continuous feeding)	2 days	Glucerna Select	48.6	NR	Nutrison Protein Plus Multifiber	35.3	NR	Insulin requirements Mean BG and GV
Vahabzadek (2019) [220]	Double-blind, parallel	No	ICU	Intervention (*n* = 44), stress hyperglycemia, 61 (53–64) years; Control (*n* = 44), stress hyperglycemia, 62 (57–64) years	Enteral feeding (nasogastric tube, orogastric tube or gastrostomy), continuous feeding with night rest	15 days	“High-fat formula”	45.0	NR	“Standard formula”	30.0	NR	Insulin requirements Capillary glucose TAG (at end of study)
Van Steen (2018) [221]	Open-label, parallel	Yes	ICU	Intervention (*n* = 52), DM2, 66.1 ± 13.2 years; Control, (*n* = 49), DM2, 67.1 ± 10.0 years	Nasogastric tube	3 days	Glucerna 1.5	45.0	29.4	Fresubin Energy Fiber	34.8	22.2	Capillary and interstitial glucose (flash monitoring) mean absolute change
Wewalka (2018) [222]	Open-label, parallel	No	ICU	Intervention (*n* = 30), 60 ± 12 years; Control (*n* = 30), 58 ± 16 years	Nasogastric tube (continuous feeding)	7 days	Diben	45.0	NR	Fresubin Original Fiber	30.0	NR	Fasting BG Diarrhea
Nourmohammadi (2017) [200]	Double-blind, parallel	No	ICU	OO + sunflower oil-based formula (*n* = 16), 58.50 ± 22.25 years; Sunflower oil-based formula (*n* = 16), 57.56 ± 23.96 years; Control (*n* = 16), 56.81 ± 24.17 years	Nasogastric tube (continuous feeding)	10 days	OO + sunflower oil-based formula/sunflower oil-based formula	45.0 /45.0	NR/NR	“Carbohydrate-based formula”	30.0	NR	HDL-cholesterol (only for OO-based formula) Diarrhea
Mesejo (2015) [223]	Open-label, parallel	Yes	ICU	Glucerna Select (*n* = 53):60 (45–71) years; Diaba HP (*n* = 52): 57 (43–70) years; Isosource Protein Fiber (*n* = 52):58 (46–68) years	Nasogastric tube	28 days	Glucerna Select/Diaba HP	49.0 /40.0	32.2/20.0	Isosource Protein Fiber	30.0	12.9	Insulin requirements Fasting BG Hypoglycemia GV
Egi (2010) [224]	Open-label, crossover	No	ICU	Intervention and control groups: stress hyperglycemia after esophagectomy (*n* = 8)	Jejunostomy, continuous feeding	16 h	“Intervention formula”	29.7	21.5	“Standard formula”	25.2	12.6	Interstitial glucose (CGM) HGI
Mesejo (2003) [225]	Single-blind, parallel	Yes	ICU	DM2 or stress hyperglycemia (*n* = 50); Intervention (*n* = 26): 65.27 ± 14.95 years; Control (*n* = 24): 64.67 ± 9.63 years	Nasgastric tube, continuous feeding	14 days	Novasource Diabet Plus	40.0	23.2	Isosource Protein	29.0	11.4	Insulin requirements Capillary and BG

ITT: intention to treat; I-formula: intervention formula; I-Fat: fat (as %) in the intervention formula; I-MUFA: MUFAs (as %) in the intervention formula; C-Formula: control formula; C-Fat: fat (as %) in the control formula; C-MUFA: MUFA (as %) in the control formula; OO: olive oil; BG: blood glucose; GV: glucose variability, HGI: high glucose index; CGM: continuous glucose monitoring; HDL: high-density lipoprotein; AUC: area under curve, NR: not reported; HbA1c: glycated hemoglobin; TAG: triglycerides.

## Data Availability

Not applicable.

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
