# Peer review of "Therapeutic Properties and Use of Extra Virgin Olive Oil in Clinical Nutrition: A Narrative Review and Literature Update"

_nutrients, 2022, doi:10.3390/nu14071440_

Round 1

Reviewer 1 Report

Materials and Methods – the word “olive oil” was not used in the search?

L 71. I would suggest removing the coma after “variety” as I think you mean to say the variety of the olives (and the ripeness).  The coma where it is now made me wonder “variety of what”?

L 113. I would suggest a reference after “anti-inflammatory activity”.

Same for the next two sentences – place the references after the specific functions.

L 127 – what are you referring when you say “these compounds”?

L 128-30. I would rephrase this sentence. I think what you are trying to say is there is no consensus on what constitutes “high phenol content”.  The wording “.. that with > 500 mg..” is particularly confusing.

L 133 consider adding “these non-oil sources”.  Or do you mean olive oil + the products?

L 137 consider adding “in the intestines” before absorption.

L 138. Consider adding “in blood” after “minor components” as I think that is what you mean, but that is not clear?

L 138/139.  Frying vegetables decreases the phenol concentration loss compared to water in ref 26, but that study did not look at baking, which is implied by the two cooking methods being in the same sentence. I don’t think you can make that comparison without citing a study that looked at both procedures.  Also, this study found the opposite:

 Xueqi, L., Bremer, G. C., Connell, K. N., Ngai, C., Pham, Q. A. T., Wang, S., Flynn, M. M., Ravetti, L., Guillaume, C., Wang, Y., Wang, S.C. Changes in chemical compositions of olive oil under different heating temperatures similar to home cooking. J Food Chem Nutr 2016; 4:1:7-15.

The difference may be olive oil exposed to air while heating v cooking vegetables (or any food) with the olive oil heating (i.e., not exposing the oil to air).

L 144-146. I found this sentence confusing.  “that decreases with ezetimibe” what are  you referring to by “that”? and then “while increases with atorvastatin” ? the absorption of B-sitosterol?

L 151.  Please add a reference after “in vivo human studies”.

L 160.  ..the therapeutic potential or capacity of these minor components,…

L 163. I am guessing the references at the end of this paragraph are for all these functions but I would consider adding after each sentence the appropriate reference as the sentences are all for different functions/ items.

L 175. Consider adding: “ .. EVOO may also reduce..”

L 179.  What is this “different mechanism”?

L 183 to 186. Please add references for the ability of olive oil to increase HDL (which a number of studies have shown) and then references for decreasing LDL oxidation.

L 239. Ref 62. This is opposite as to what was found in:

Ryan M, McInerney D, Owens D, Collins P, Johnson A, Tomkin GH. Diabetes and the Mediterranean diet: a beneficial effect of oleic acid on insulin sensitivity, adipocyte glucose transport and endothelium-dependent vasoreactivity. Qjm 2000;93:85-91.

L 277. Please provide the total phenol content for the “polyphenol-rich EVoo” cited.

L 285. What was the comparison to ?

L 289. NHANES uses the Med diet Score which would not indicate EVOO consumption in the American diet.

L 301. Please provide the amount of EVOO supplementation.

L 302. What were the food sources in the diet that made it polyphenol-rich? Was EVOO included?

L 304. Please provide the total phenol content of the polyphenol-rich EVOO.

L 322. What was the comparison group?

L 327. What was the comparison group?

L 338.  High MFA diets – did any or all of these studies use EVOO (or olive oil)?

L 356 “… MFA intake and has not changed..” there seems to be a word(s) missing before “has”?

L 366 “at the expense” do you mean “due to” or “because of”

L 371 consider adding “..rich in oleic acid and other MFAs” (as oleic acid is an MFA)

L 384 “In our review”?

L 425. My understanding is that there is sufficient 18:2 in olive oil (also shown in your Table 1) to avoid an essential fatty acid deficiency is only olive oil is consumed and not vegetable seed oils. If there is evidence to suggest that using olive oil alone could cause an essential fatty acid deficiency that should be included.

L 431. What would be examples of the “differences” you cite? anything noteworthy?

L 471. Was the amount of hydroxytyrosol used in both studies the same? If not, that may at least partially account for the different results (besides time).

L 477. Can the phenolic compounds increase interleukin-6 to levels that would provide anti-obesogenic properties?

L 560. Please include the amount of calcium supplemented.

L 573. Consider changing “at 16 weeks” to “by 16 weeks” as I think you mean although the study was 7 years, the effect was seen earlier.

Author Response

Dear colleague, 

Thank you very much and best wishes,

Andrés

Reviewer 2 Report

This review by Andrés Jiménez Sánchez and collaborators reports a very nice and useful summary on the most relevant therapeutic properties of Extra Virgin Olive Oil (EVOO) in clinical nutrition. The paper is very well written and the information reported covers the majoritiy of properties of EVOO and its components. The paper is organized in sections. Data referring to the Spanish EVOO are also presented in tabellar form, which makes it useful not only as a general report but also as a report on the Spanish olive oil.

Major points

None

Minor points

- Please make it clear that the term 'population/s' used throughout the text means 'clinical populations' in certain contexts. Otherwise it could be misleading. See e.g. line 312 or 446.

- While I have no objections on the manuscript organization and content, there is a need to revise the 'References' section thoroughly to make it uniform in format.

Author Response

(The authors gave the same response as above.)

Reviewer 3 Report

To the Authors (in detail):

  1. The argument is interesting but has to be improved in some sections. The first part of the manuscript has to be widely improved because the discussion of the olive oil composition is not complete, in fact some component is not discussed and other components are poorly discussed. You have related your work mainly on studies of Spanish olive oil (see tables) even if the title is not related only to a Spanish context. For this reason the bibliography has to be improved and extended with reference regarding EVOO of different geographical areas of the olive oil production. Some component of the EVOO is not treated (fatty alcohols/linear alcohols/policosanol and n-alkanes and n-alkenes) other important compounds are poorly treated (sterols). There are inaccuracies in the text, including in the references section. Table 1 do not contain all fatty acids of an EVOO, in particular are missed: myristic, behenic and lignoceric acids whose content is required by the European regulation and by the International Olive Oil Council. The second part of the manuscript is better treated, in particular with regard the effects of FAMEs and phenols on the human health.

  1. Introduction section, line 43, replace 0.8 º with 0.8%; in addition , EVOO has an acidity ≤ 0.8% and not less than 0.8%;

  1. Introduction section lines 42-44. To classify an olive oil as EVOO it has to respect all the parameters listed in the European Regulation on olive oil and the Trade Standard edited by the International Olive Council. The acidity olive oil has an acidity ≤ 0.8%, this is not enough to declare it as EVOO. Please, read the regulations and re-arrange this sentence;

  1. Introduction section, lines 42-44, the references you have listed (1-7) are not really proper to support your statement because they are not the origin of the chemical datum (0.8%) but only report the original datum. I suggest to replace two of them or all 7 with the following references which are the European Regulation on olive oil [1]  and the Trade Standard edited by the International Olive Council [2];

[1] Commission Regulation (EEC) No 2568/91of 11 July 1991 on the characteristics of olive oil and olive-residue oil and on the relevant methods of analysis. (OJ L 248, 5.9.1991, p. 1). Consolidated Text. 01991R2568 — EN — 04.12.2016 — 031.005 — 1

[2]  International Olive Council. Trade Standard applying to olive oils and olive pomace oils. COI/T.15/NC No 3/Rev. 12, June 2018, English version, Original: French version.

  1. 1.1 sub-sub-section and in the whole manuscript, verify if to use the term variety or cultivar. In the case of olive I suggest to use cultivar acronyms of cultivated variety. Olea europaea is a cultivated plant. In fact in your table 1 you have used cultivar ad not variety. Verify the whole manuscript;

  1. Table 1 is not complete: C14:0, C22:0 and C24: are missed;

  1. Lines 97-98: insert one proper citation for less than 250 minor component (please use an International relevant journal). In addition they are less than 250 minor component or less than 250 components? Please verify and include 1-2 references;

  1. Line 106, after the dot you have to write β-Tocopherol, T in capital letter;

  1. Lines 108-109, this is not true: 95% (better >93%) is the so called apparent β-sitosterol which is the sum of five sterols (see in my list [1, 2], please verify and re-arrange this sentence. β-sitosterol apparent is the parameter required by [1, 2]. Also if you want to write only about specific β-sitosterol, you have to re-arrange this sentence and to include proper bibliography;

  1. 1.1 sub-sub section, you have completely missed other components such as fatty alcohols which are very important in the human diet. Please, extend and complete this section and find, read and discuss the following references [3-4]; After this, discuss the importance of fatty alcohols in your 3.1.5. section and support the discussion with the following references [5-6-7]:

[3] The effects of cultivar and harvest year on the fatty alcohol composition of olive oils from Southwest Calabria (Italy).

Grasas y Aceites 65: e011 (2014).  http://dx.doi.org/10.3989/gya.073913

[4] Policosanol variation in olive oil as a result of variety, ripening, and storage.

Eur. J. Lipid Sci. Technol. 2015,117, 1248–1260. DOI: 10.1002/ejlt.201400483

[5] Taylor JC, Rapport L, Lockwood GB. 2003. Octacosanol in human health. Nutrition 19, 192–195.

[6] Clinical and Experimental Pharmacology and Physiology (2002) 29, 891–897

ANTIPLATELET EFFECTS OF POLICOSANOL (20 AND 40 MG/DAY) IN

HEALTHY VOLUNTEERS AND DYSLIPIDAEMIC PATIENTS.

Int J Clin Pharmacol Ther. 1998 Sep;36(9):469-73.

[7] Long-term therapy with policosanol improves treadmill exercise-ECG testing performance of coronary heart disease patients

Int J Clin Pharmacol Ther. 1998 Sep;36(9):469-73.

  1. 1.1 sub-sub section, you have completely missed other components such as n-alkanes and n-alkenes. Please, extend and complete this section and find, read and discuss the following references [8-9-10]; After this, discuss the importance of n-alkanes and n-alkenes in your sub-sub-section 3.1.5, at least with the with the following reference [11]:

[8] The effect of cultivar and harvest season on the n-alkane and the n-alkene composition of virgin olive oil.

European Food Research and Technology 247 (1) 25-36 (2021).

doi: 10.1007/s00217-020-03604-x

[9] n-Alkanes and n-alkenes in virgin olive oil from Calabria (South Italy): the effects of cultivar and harvest date.

Foods 2021, 10 (2) 290. Special Issue "New Insights into Specificity, Authenticity and Traceability Analysis of Olive Oils". https://doi.org/10.3390/foods10020290

[10] Occurrence of n-Alkanes in Vegetable Oils and Their Analytical Determination

Occurrence of n-Alkanes in Vegetable Oils and Their Analytical Determination

[11] The European Agency for the Evaluation of Medicinal Products. Committee for Veterinary Medicinal Products—Mineral Hydrocarbons—Summary Report; EMEA/CVMP/069/95-Final; European Food Safety Authority: Parma, Italy, 1995.

  1. 3.1.1 sub-sub section, when you discuss about volatiles in olive oil, explain, evidence the factors influencing the composition [12] and that cooking process modify their composition [13] as you made for phenols. Extend the discussion and find read and discuss also:

[12] Evolution of Flavors in Extra Virgin Olive Oil Shelf-Life.

Antioxidants 2021, 10, 368. https://doi.org/10.3390/antiox10030368

[13] Volatile profiles of extra virgin olive oil, olive pomace oil, soybean oil and palm oil in different heating conditions.

LWT - Food Science and Technology 117(1), 108631 (2020).

https://doi.org/10.1016/j.lwt.2019.108631

  1. 3.1.2. sub-sub-section,  lines 124-126. Many other factors influence olive oil quality: geographical area of production, agronomic production factors, harvest year, harvest system, olive oil extraction system, storage method. Please extend the discussion and the bibliography. I suggest two references but you have to find some reference more [14-15]:

[14] EFFECT OF EXTRACTION SYSTEMS ON QUALITY CHARACTERISTICS OF EXTRA VIRGIN OLIVE OIL. Arab Univ. J. Agric. Sci., Ain Shams Univ., Cairo, Egypt 27(4), 2167-2176, 2019

[15] Virgin Olive Oils: Environmental Conditions, Agronomical Factors and Processing Technology Affecting the Chemistry of Flavor Profile

Journal of Food Chemistry and Nanotechnology | Volume 2 Issue 1, 21-21 (2016).

  1. 3.1.3. sub-sub-section. Yes, phenolic content decrease during cooking. Many studies were conducted in this filed but you have included only one. Please, extend this section, find, read and discuss at least [16-17].

[16] Effect of heating on chemical parameters of extra virgin olive oil, pomace olive oil, soybean oil and palm oil.

Italian Journal of Food Science, Volume 30, Issue 4, Pages 715 – 739, 30 October 2018

[17] Microwave Heating Induces Oxidative Degradation of Extra Virgin Olive Oil

Food Science and Technology Research, 25 (1), 75_ 79, 2019.  doi: 10.3136/fstr.25.75

  1. 3.1.3. sub-sub-section. You have only mentioned baking and its implications but you have not discussed this point. EVOO is included in the recipe to improve the nutritional properties of a bakery product. Please extend this section and support your statement with proper references. Please, find, read and discuss: [18, 19, 20].

[18] Potential use of extra virgin olive oil in bakery products rich in fats: A comparative study with refined oils. Int. J. Food Sci. Technol. 2013, 48, 82–88

[19] Effects of shortening replacement with extra virgin olive oil on the physical–chemical–sensory properties of Italian Cantuccini biscuits.

Foods 2022, 11, 299. https://doi.org/10.3390/foods11030299

[20] Olive oil by-product as functional ingredient in bakery products. Influence

of processing and evaluation of biological effects

Food Research International 131 (2020) 108940. https://doi.org/10.1016/j.foodres.2019.108940

  1. Sterols. Your discussion about sterols is poor, please extend the discussion about the sterol composition, the importance of sterols in the human diet. Please, find read and discuss the following papers [21, 22, 23, 24, 25]:

[21] Plant sterol consumption frequency affects plasma lipid levels and cholesterol kinetics in humans.

European Journal of Clinical Nutrition (2009) 63, 747–755

[22] Intake of dietary plant sterols is inversely related to serum cholesterol concentration in men and women in the EPIC Norfolk population: a cross-sectional study.

European Journal of Clinical Nutrition (2004) 58, 1378–1385

[23] Biological effects of oxidized phytosterols: A review of the current knowledge.

Progress in Lipid Research 47 (2008) 37–49

[24] Progress in Lipid Research 47 (2008) 37–49. Progress in Lipid research 50 (2011) 357-371. doi:10.1016/j.plipres.2011.06.002

[25] The Story of Beta-sitosterol- A Review. European Journal of Medicinal Plants 4(5): 590-609, 2014

  1. References section, the journal name has to be abbreviated: see your refs:  62, 71, 80, 94, 110, 132, 134, 189, 191, 193, 202, 216, 230, 232. Please, verify also other your references;
  2. References section, sometime you have written the title of the paper in capital letters (ref 13) and sometime in small letters (ref 12). Please, be consistent in the whole section;
  3. References section: the scientific names in italic and the species in small letters: Olea europaea and not Olea Europea, see your refs: 19, 29, 44;
  4. References section and in the whole manuscript, verify how to have written the title of the paper and do not write (et al) but be consistent with the instructions for authors of Nutrients;
  5. Please, evidence your corrections.

In my opinion, a major revision is necessary.

Regards.

Author Response

(The authors gave the same response as above.)

Round 2

Reviewer 3 Report

For Authors

  1. Page 3, lines 129, 135 and in the whole manuscript and tables; when you write the names of sterols, decide if to write the position as an exponent or not:  Δ5,24-stigmastadienol or Δ5,24-stigmastadienol? The same for each sterol, verify the whole manuscript;
  2. Page 3 and in the whole manuscript, sometime you have written β-sitosterol (s in small letter) and sometime β-Sitosterol (S in capital letter), the same for all sterols. Please, verify and be consistent: capital or small letter?
  3. Page 3 and in the whole manuscript, decide if to write the name of all molecules (compounds) in capital or small letters. This not only for sterols but for all molecules in your manuscript;
  4. Lines 176, 237 and in the whole manuscript: when you write a temperature, separate the numeric value from the symbol: 20 °C and not 20°C;
  5. References section: the references have to be reported as required by Nutrients (MDPI). For example: the publication year in bold; The journal name italicized and abbreviated (ref 16); the volume number italicized;
  6. References section, please, verify each title of paper: ref 16, in capital letter, other references in small letters. I have indicated some reference as an example, but the whole section has to be arranged as required by Nutrients. Please, read the instructions for authors;
  7. Please, evidence your corrections.

In my opinion, a minor revision is necessary.

Regards.

Author Response

(The authors gave the same response as above.)
